# Treatment of Lower Risk Myelodysplastic Syndromes

Valeria Santini 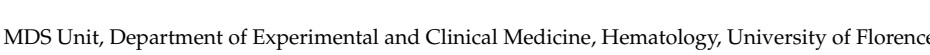

MDS Unit, Department of Experimental and Clinical Medicine, Hematology, University of Florence, Largo Brambilla 3, 50134 Firenze, Italy; valeria.santini@unifi.it; Tel.: +39-055-794-6647

**Abstract:** Purpose of review: Management and Optimization of therapy for lower-risk myelodysplastic syndromes will be reviewed here. Recent findings: Lower-risk MDS typically present with clinical manifestations of anemia, which is the most frequently encountered cytopenia in this setting. While therapy with erythropoietic stimulating agents (ESAs) is used in the vast majority of cases, if correctly selected, some patients do not respond, or become irresponsive to ESAs. Novel agents with very different modes of action show promising clinical results in anemic LR-MDS refractory/relapsed after ESAs. Luspatercept, a TGFbeta family ligand trap, induces nearly 50% of transfusion independence in LR MDS. Another investigational agent showing efficacy and possibly disease modifying activity is the telomerase inhibitor imetelstat. Modulation of dose and schedule of hypomethylating agents, both injectable and oral, is currently being explored, and preliminary results are positive. There is still no standard therapeutic approach for thrombocytopenic and neutropenic LR MDS, although they do represent a smaller proportion of cases. Immunosuppressive treatments, as well as TPO mimetics, could represent a good option in selected MDS cases. Summary: At present, the availability of novel active agents allows the planning of sequential therapy, especially for anemic LR MDS. Better diagnosis and prognostic stratification may allow a more precise and personalized treatment.

**Keywords:** myelodysplastic syndromes; IPSS-R lower risk; erythropoietic stimulating agents; target therapy

## 1. Introduction

Definition of "*lower-risk*" myelodysplastic syndromes (MDS) constitutes a challenge. According to the revised international prognostic scoring system (IPSS-R), we may indeed distinguish subgroups of MDS categorized within the very low, low, and intermediate (but the latter with a score of <3.5) categories that, as a whole, present survival longer than 1.5 years. Nevertheless, although useful, IPSS-R has shown limitations in stratification. Patients with isolated neutropenia or thrombocytopenia can pose problems, first in diagnosis, but also in terms of prognostication. Thus, although we refer to IPSS-R categories (and still IPSS ones) to prescribe treatments, we have to acknowledge that there is a further level of granularity in prognostic variables and that we should indeed individualize even more our evaluation of risk. In this sense, the implementation in MDS diagnostic apparatus of somatic mutation evaluation, as recently proposed, certainly improves this assessment. Integration of clinical and genomic characteristics has led to the proposal of a new MDS classification [1].

The molecular international working group (IWG) also has implemented genomic and clinical data, producing a scoring system based on the evaluation of recurrent mutations, hemoglobin (Hb), bone marrow blast, and platelets that re-positions cases in six risk categories, shifting several IPSS-R low risk cases towards IPSS-M higher risk [2].

## 2. Therapy

Anemia with hemoglobin levels <10 g/dL affect 50% of the MDS cases included in the Italian MDS Registry (data on file FISIM www.fisimematologia.it, accessed on 1 January 2022). They impact on symptoms and quality of life, which is particularly evident

for LR MDS patients whose survival is longer and who often become transfusion dependent. Consequences of chronic anemia in terms of cardiac remodeling are thus frequent. In fact, the incidence of cardiac remodeling in transfusion-dependent vs. transfusion-independent patients was demonstrated to be 92% vs. 48%, respectively. For every 1 g/dL increase in Hb, there is a predicted 49% decrease in the risk of remodeling ($p = 0.004$) [3].

Transfusions often remain the first approach to treat severe anemia and sometimes are the only option offered and immediately available for LR MDS patients [4]. Especially for patients with a high transfusion burden, frequent accesses to hospital and dependence on caregivers decreases quality of life, together with oscillation in Hb values that determine the evident periodical recurrence of symptoms. The hemoglobin threshold for transfusions is generally 8 g/dL in the majority of European Countries, but of course this threshold is different for patients with recognized cardiac problems or who are reporting severe symptoms such as dyspnoea and excessive fatigue. Moreover, the target levels of Hb after transfusion may vary but are usually around 9 g/dL and do not allow a complete management of symptoms. All these considerations, together with the complications related to chronic transfusions, such as alloimmunization and iron overload, should discourage the exclusive use of this supportive treatment without attempting other therapeutic approaches.

In case of lack of alternative therapies, the transfusion regimen should be accompanied by an optimal support of iron chelation. There has been long and lively debate around the opportunity and timing of iron chelation in LR MDS patients, but evidence has accumulated indicating the relevance of organ damage by excessive iron intake and its avoidance by optimal chelation. While national and international guidelines have not always been consistent in their recommendations [4], the final data in this sense were obtained by a randomized prospective international clinical study comparing deferasirox therapy versus placebo in transfusion-dependent LR MDS patients [5]. This study demonstrates that, for chelated patients, there was a 34% decreased risk of serious/fatal events and prolongation of survival of 398 days when compared with placebo patients [5].

The best management of anemic LR MDS implies an early treatment with erythropoietic stimulating agents (ESAs) before the onset of transfusion dependence or within 6 months from its inception. Prompt treatment with ESAs is in fact inducing higher response rates [6]. ESAs have been used successfully for decades to ameliorate anemia of LR MDS, but were only recently approved, and only in Europe [7].

The registration clinical trial for erythropoietin alpha that finally led to its approval shows a response rate inferior to what we are used to observing and what is reported in "real life" use. In fact, only 31.8% of treated patients had a response, vs. 4.4% of placebo ones [8]. This outcome is due to the design of the study, which provoked problems in the evaluation of its outcome, and difficulties in applying IWG criteria for frequent suspension of treatment when Hb level was approaching 12 g/dL, together with the inclusion of numerous transfusion dependent patients and the use of a weight adjusted dose of erythropoietin alpha instead of the standard fixed dose [8]. Selection of patients is critical to obtain responses. As already mentioned, long-standing transfusion dependence, >5% bone marrow blasts, a complex karyotype, and high ferritin levels, as well as multilineage dysplasia and IPSS-R higher risk predict poorer responses [9,10], but the fundamental variable is the level of endogenous erythropoietin, which should be <200 U/L to obtain optimal an response rate. In multiple real-life studies, the actual rate of response is in fact around 60% for the patients selected, according to the above criteria. Regarding the doses, it is generally accepted that the standard dose of ESAs is 30–40 U subcutaneously/week, while higher doses have been proposed, mainly by Italian researchers [11], without a significant impact on response, duration of response, or overall survival, apart for the group of patients with Hb 8–10 g/dL, with a diagnosis of MDS with unilineage erythroid dysplasia, MDS-RS, or del(5q) [12]. Interruption of treatment with ESAs almost always provokes loss of response. It has been suggested that patients responding to ESAs may have prolonged OS.

At present, no specific somatic mutation has been linked to the propensity of response to ESAs, while the higher number of mutations is associated only with a trend towards a lower response, but in multivariate analysis it correlates with a significantly shorter survival despite ESA treatment [13].

The widespread use of biosimilar erythropoietic agents seems to yield the same rate of response in treated patients [14]. Thrombotic events are rare, provided Hb levels are controlled. In fact, ESAs agents are not associated with increased risk of thrombosis in patients with MDS, as demonstrated by an early study in which deep vein thrombosis (212 cases/5673 MDS) occurred only in patients with a central venous catheter and TD [15].

Although very effective if prescribed to selected patients, ESAs show transient activity, and patients often become chronically transfusion dependent.

Several novel agents are under evaluation in the setting of ESA relapsed/refractory LR MDS patients (Figure 1).

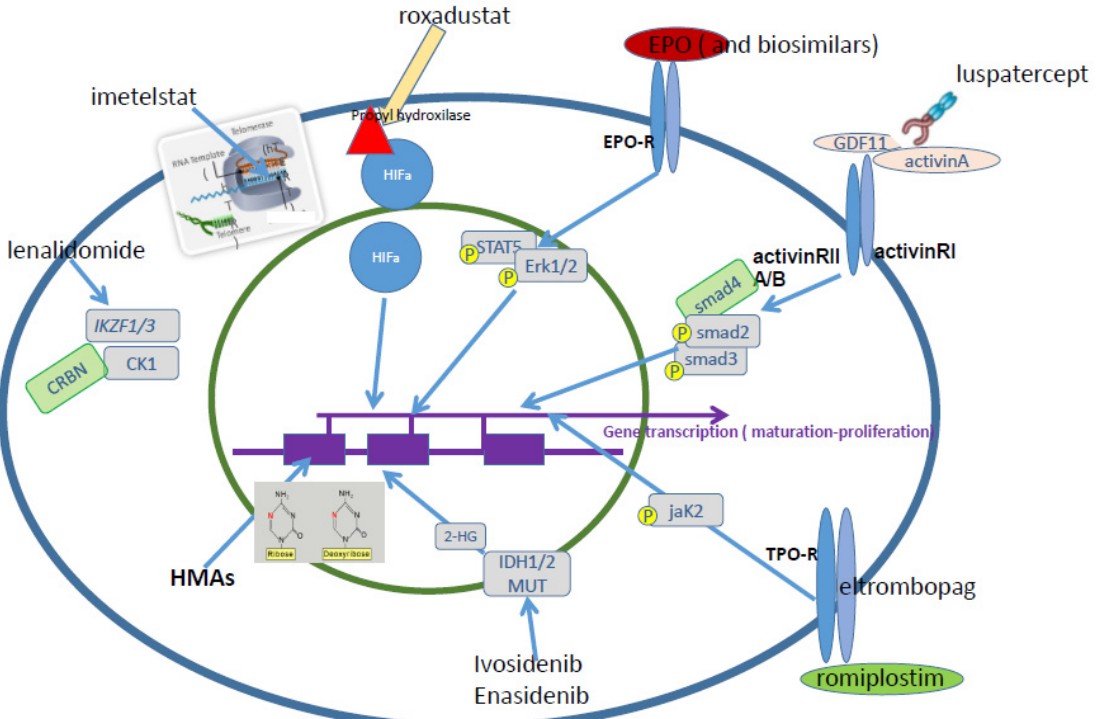

**Figure 1. Possible therapeutic agents in LR MDS.** Erythropoietin alpha; Imetelstat—competitive telomerase inhibitor-Proprietary 13-mer thio-phosphoramidate (NPS) oligonucleotide, with covalently-bound lipid tail to increase cell permeability/tissue distribution; Roxadustat—Oral hypoxia-inducible factor (HIF) prolyl-hydroxylase inhibitor, stabilizes HIFalpha; HMAs—hypomethylating agents (decitabine and azacitidine). Induce decrease of methylated cytosines in selective regions of DNA; Lenalidomide mediates the ubiquitination and degradation of Ikaros family zinc finger protein 1 (IKZF1), IKZF3, and casein kinase 1α (CK1α) by facilitating their interaction with cereblon (CRBN); Ivosidenib—oral selective inhibitor of metabolic enzyme isocitrate dehydrogenase 1 (IDH1) mutated form, decreasing 2-HG levels; Enasidenib—oral selective inhibitor of metabolic enzyme isocitrate dehydrogenase 2 (IDH2) mutated form, decreasing 2-HG levels; Eltrombopag—oral small molecule, binds at a trans-membrane site of TPO-R; Romiplostim—subcutaneous peptibody, binds directly and competitively at the TPO binding site of TPO-R.

Chronic anemia typical of MDS is accompanied and worsened by inflammation. Production of cytokines such as IL-6, TGFbeta, TNFalpha, and SA1008/SA1009 produced by mesenchimal cells or other cells from the erythropoietic niche is known to aggravate anemia in MDS [16,17].

While some attempts to inhibit TGFbeta and TNFalpha cytokines have been made without great success in the past [18], more recently, TGF-β pathway inhibitors have been investigated: The TGF-β receptor I kinase/ALK5 oral inhibitor galunisertib demonstrated activity with 44% hematological improvement in TD LR MDS, without any specific MDS subgroup showing increased sensitivity [19]. Sotatercept and luspatercept are activin receptor ligand traps that have been employed in clinical trials in LR MDS [20,21]. The erythopoietic stimulating activity of sotatercept was a serendipity finding in female breast cancer patients treated with that drug to increase bone density [22]. Among the two agents, luspatercept, activin RIIB/IgG1 Fc recombinant fusion protein, by trapping ligands such as GDF11 and others, has the highest erythropoietic stimulating activity. It allows restoration of terminal maturation of erythroid progenitors by diminishing SMAD2/3 signaling. After the positive phase II study [21], indicating a significantly higher propensity to respond in MDS with ring sideroblasts (MDS-RS), the international phase III randomized study demonstrated that luspatercept, administered subcutaneously every 3 weeks, induces transfusion independence (TI) in nearly 50% of MDS-RS cases, especially if transfusion burden does not exceed 6 U RBC/8 weeks [23], with negligeable side effects and with a cumulative duration of RBC-TI $\geq$ 8 weeks (sum of all periods of RBC-TI $\geq$ 8 weeks) of 79.9 weeks (53.7–112.3) vs. 21.0 for the placebo controls. Luspatercept is approved by the EMA and FDA for TD LR-MDS-RS patients who are refractory or resistant to ESAs, as well as for TD beta-thalassemia. Clinical trials are ongoing to evaluate luspatercept efficacy in non-RS MDS in first line therapy vs. ESAs, as well as in other studies regarding different therapies in combination with ESAs.

Lenalidomide, approved for treatment of MDS-del5q, is an immunomodulating agent inducing transfusion independence in these specific MDS subtype, acting via ubiquitination and degradation of Casein Kinase by the E3 ubiquitin ligase [24,25]. A pivotal study indicated that up to 83% of patients with TD MDS del5q obtain an erythroid response with lenalidomide therapy [26]. More stringent criteria of response, applied in the last placebo-controlled study, demonstrated that patients with RBC-TD MDS del5q (*n* = 205) achieve RBC-TI $\geq$ 26 weeks in 43–56% of cases. The optimal dose for TI and cytogenetic complete response (CCyR) is 10 mg/day [27]. On the other hand, Lenalidomide-induced CCyR is lower in TP53 mutated patients, and mutation of TP53 was demonstrated to predict poor outcome and progression in MDS del5q [28]. This observation was corrected by a recent work, showing that it is the TP53 allelic state that shapes clinical outcomes, with multi-hit, biallelic TP53 mutations impacting on response and survival after lenalidomide, while patients with monoallelic mutations of TP53 have a survival time similar to those who carry wild type TP53 [29]. A recent study, in an attempt to evaluate early treatment onset with lenalidomide, demonstrated that in non-TD MDS-del5q, low doses of the drug (5 mg/daily for 21 days) may delay and decrease TD. The time to transfusion dependence was 6.3 years for lenalidomide-treated patients vs. 2.8 years, with a 61.2% decrease in the risk of TD [30].

In a SEER Medicare database evaluation, among 676 lenalidomide treated patients, 35% were non-del(5q) MDS and 43% were MDS-U, despite indications for prescription of the drug. Patients with LR MDS who fail ESAs or relapse after initial therapy are a group of patients for whom there are very limited options, and this is the reason why off-label lenalidomide is prescribed so often [31]. We demonstrated in a phase III randomized trial vs. placebo that lenalidomide may induce TI > 8 weeks in 27% of TD LR non-del5q MDS. The clinical indicator of a higher probability to achieve TI was endogenous erythropoietin levels < 100 U/L (42.5% TI > 8 weeks), but we could not characterize other specific predictive factors for response in terms of gene expression signature or specific mutation profile [32,33]. As a matter of fact, we observed a trend for lower response rates in patients carrying the ASXL1 mutation, and higher response rates for the ones with mutated DNMT3a and EZH2, the limited number of cases being an obstacle to achieving statistically sound considerations. As expected, a higher number of mutations was significantly associated with a shorter median OS (*p* = 0.0005) and DNMT3A mutant patients had a trend for improved OS

with lenalidomide treatment compared with placebo ($p$ = 0.123) [33]. The combination of lenalidomide and ESAs yields 38.9% of major erythroid responses in TD non-del5q LR MDS patients who relapsed after ESAs or had a low probability to respond to ESAs [34].

Imetelstat, a telomerase inhibitor that is administered intravenously every 4 weeks, induced TI in 42% of treated LR MDS that were refractory/resistant to ESAs and were not pre-treated with lenalidomide or hypomethylating agents [35]. This activity of imetelstat is accompanied by an on-target effect, which is a >50% reduction in hTERT expression [36] and by the decrease of mutational burden in patients responding to treatment (i.e., SF3B1 mutants). Response duration may be >1 year, as shown in around 30% of cases, and as long as 2.8 years [35]. A phase III randomized clinical study comparing imetelstat to placebo has just been completed, and the results are awaited.

Another agent under investigation in LR MDS is Roxadustat, an oral HIFalpha hydroxylase inhibitor, recently approved by the EMA in China and Japan for anemia of renal insufficiency. The results of this therapy in low burden TD LR-MDS are promising: 38% of patients achieved TI > 8 weeks in weeks 1–28 of treatment, while 42% achieved this objective during 52 weeks of treatment. A reduction of >50% in transfusion burden was observed in 54–58% of cases. [37,38].

Hypomethylating agents (HMAs) in LR MDS, especially when multilineage cytopenias are present [39], may be a good therapeutical option, although their use is not approved for LR MDS in Europe. Modulation of HMA doses and schedule have recently been proposed. Attenuated doses of decitabine 20 mg/m$^2$/3 days induced 67% TI in LR MDS, while azacitidine 75 mg/m$^2$/day for 3 days had an overall response rate of 49% and a TI of 16% [40].

While it has been known for many years that injectable HMAs at standard doses are very active in promoting hematological improvements in LR MDS [41], it is only recently that oral formulations have been evaluated in this setting. In 2020, the FDA and Canadian authorities approved decitabine/cedazuridine (ASTX727 or DEC-C, oral decitabine) for the treatment of all subtypes of adult MDS and CMML in any stage of the disease. The approval is based on the evidence that the fixed dose oral cedazuridine/decitabine (100 mg/35 mg) has systemic exposure and DNA demethylation and safety is equal to decitabine 20 mg/m$^2$/day intravenously for 5 days, with similar efficacy (60% overall response rate) [42]. In the Ascertain trial, 69/133 MDS patients treated with DAC-C had LR MDS. Their overall response rate (ORR; CR + PR+ mCR + HI) was 57%. Among the patients who were RBC or platelets TD at baseline, 48% became RBC-TI and 67% became platelets TI [43].

The oral azacitidine CC-486 was recently approved by the FDA for maintenance treatment of AML in first remission following induction chemotherapy [44]. This agent differs from injectable azacitidine due to its pharmacokinetic or pharmacodynamic profile. CC-486 may allow for extended dosing schedules over each treatment cycle and prolong drug exposure. We recently showed that CC486, given for 21 days 300 mg/daily, induced significant transfusion independence and durable bilineage improvements in TD LR-MDS with thrombocytopenia and high-risk features [45]. Fatal infections in patients with severe baseline neutropenia were observed, warranting further evaluation of the role of CC 486 in LR MDS [45].

These clinical results, indicating high efficacy of oral HMAs in LR MDS, pave the way towards a total oral therapy for this disease.

Although quite rarely (<10% of cases), MDS patients may carry somatic mutations of IDH1 or IDH2 genes, which are well targetable by the specific inhibitors of the mutants, ivosidenib and enasidenib, oral agents approved for the therapy of R/R acute myeloid leukemia but not yet for MDS. Nevertheless, their efficacy in MDS has already been demonstrated, specifically for higher-risk MDS cases [46,47]. Target therapy in LR MDS with ivosidenib and enasidenib may be a possible option for patients with ESA refractory or relapsed anemia, especially for those with additional cytopenias. Recent data from clinical studies of the Groupe Francophone des Myelodysplasies, although preliminary, indicate

the activity of both drugs. In TD post ESA failure LR MDS carrying the IDH1 mutation, with a median VAF of 44%, response rate to ivosidenib 500 mg/day was 50% [48]. The administration of enasidenib 100 mg/day in the same type of LR MDS patients but with the IDH2 mutation, median VAF 36%, equally led to a 50% response [49].

In a recent survey of MDS cases in the Italian MDS Registry, among 6819 cases, only 37% presented with a number of platelets <100,000/microL. Notwithstanding the fact that it is a relatively rare finding, the presence of thrombocytopenia impacts negatively and significantly on MDS patient prognosis. Therefore, thrombopoietin (TPO) mimetics have been extensively evaluated in thrombocytopenic MDS. There are two main agents available in clinics, both approved for the therapy of immune thrombocytopenia and with different mechanisms of action and route of administration. Neither has been approved for MDS, although they have been suggested regarding this diseases as therapeutic options by several Authors [39,50]. Eltrombopag is an oral TPO receptor agonist interacting with the transmembrane domain of c-Mpl. Its activity has been shown in two phase 2 open-label studies of LR MDS. In patients with severe thrombocytopenia (platelets $< 30 \times 10^9$/L), at doses ranging from 50 to 300 mg/day, eltrombopag induces a significant increase in platelet number in a substantial proportion of cases (47% vs. 3% placebo) and provokes a decrease in platelet transfusions, as well as in hemorragic events [51]. A similar response rate was observed in the second phase 2 dose escalation study, where 44% of patients responded. The predictors of response were the presence of a PNH clone, marrow hypocellularity, thrombocytopenia, and baseline elevated plasma TPO levels [52].

Romiplostim is a subcutaneous agent constituted by a fusion peptibody TPO analog that increases platelet production via binding and activation of the thrombopoietin (TPO) receptor (c-Mpl). The phase 2 studies of this agent yielded results similar to those presented for eltrombopag: at doses of 750 microg weekly/biweekly, romiplostim induced a 46% increase in platelets and a decrease in severe bleeding events and platelet transfusions [53]. In a subsequent randomized trial, platelet responses were observed in 36.5% of LR MDS patients treated with romiplostim [54]. The increase in marrow blasts that blocked the phase 2 study, after re-evaluation with long follow up, indicate that this is a transient effect of the drug and not progression to acute leukemia [55].

There is accumulating evidence that, in MDS, immune alterations have an important pathogenetic role. Similarly to what is observed in aplastic anemia, stimulation of autoimmunity may be the basis of some MDS; as a matter of fact, aplastic anemia, paroximal nocturnal hemoglobinuria, and MDS share common features and clinically overlap quite significantly. The positive response to immunosuppressive therapy supports this notion. Notwithstanding a reported cumulative overall response rate of 48.8% with 30% CR after all types of immunosuppressive treatments (alemtuzumab, cyclosporine, etanercept, horse ATG and rabbit ATG, sirolimus), this therapeutic approach is underutilized in MDS [56]. A recent meta-analysis of >500 MDS patients treated in 13 different clinical trials confirmed a cumulative response in 42.5% of cases [56]. The most frequently used regimen involves anti-thymocyte globulins (ATG), with or without cyclosporine. In the FISiM Registry, of >2000 MDS cases analyzed, only 1.5% received immunosuppressive treatment as a first-line therapy (Attardi et al., manuscript submitted).

Over the years, the evaluation of sparse data in heterogenous studies with unselected patient populations has allowed the characterization of some predictive factors of sensitivity to immunsuppressants: initial erythroid response, hypocellular bone marrow (age adjusted or cellularity <20%), BM blast count <5%, use of horse ATG plus cyclosporine versus rabbit ATG or ATG without cyclosporine [56]. The literature is not consistent regarding the predictive role of age, transfusion dependence, Paroxysmal nocturnal hemoglobinuria, T-LGL clones, or HLA DR15 positivity. In contrast, several years ago, the following formula to select MDS patients for immunosuppressive therapy was proposed by the NIH group: HLA-DR15–patient: age + months of RBC transfusion dependence (RCTD) ≤ 58; HLA-DR15+ patients: age + months of RBC transfusion dependence (RCTD) ≤ 72 [57]. Given the numerous discrepancies, it would be advisable to perform prospective randomized studies.

Increasing evidence indicates that innate immune activation and excessive generation of inflammatory proteins, such as tumor necrosis factor-α (TNFα), interleukin (IL)-1β, IL-6, alarmins, among others, both play a role in the pathophysiology of MDS. Chronic inflammation, as observed during aging, is also present in LR MDS [58]. Preclinical observations suggest that IL1/Toll-like receptor signaling and IRAK4 inhibition may improve hematopoiesis in murine MDFS models [59].

An absolute neutrophil count <800/microL is a very rare finding in MDS: only 17% of MDS cases in the Italian Registry show such a level of cytopenia. We demonstrated that MDS patients with neutropenia have generally a mild course and a good prognosis [60]. In line with this observations, the recent proposal of the IPSS molecular scoring system does not include neutropenia <800/microL in the variables to calculate the score due to lack of prognostic significance [2]. There are no specific therapies aimed at targeting isolated neutropenia, with an absence of evidence for a role of filgrastim in prophylaxis in cases without infective events.

Hematopoietic stem cell transplant (HSCT) is the only curative option for MDS at present. Generally, lower risk MDS patients are not considered candidate for HSCT. Notwithstanding better prognostic stratification, it is still rather challenging to indicate the best time to transplant. Regarding this point, a study that analyzed a large number of MDS cases was published several years ago and suggested that allogeneic HSCT offers optimal survival benefits when the procedure is performed before progression to more advanced disease, in particular until progression to intermediate-1 IPSS risk [61].

The most recent recommendations for HSCT in MDS indicate that younger and fitter LR MDS may be considered for a curative approach with HSCT if they present with frequent RBC transfusions (≥2 units per month), life-threatening cytopenias (neutrophil counts <0.3 × 109/L or platelet counts <30 × 109/L), and very poor prognostic cytogenetic markers. The introduction of molecular parameters in prognostic stratification may support further discrimination and allow better identification of LR MDS with worse prognosis, and thus identify potential candidates for early HSCT [62].

**Funding:** This research received no external funding.

**Conflicts of Interest:** Advisory board for BMS/Celgene, Geron, Gilead, Menarini, Novartis, Takeda.

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
