# Peer review of "Treatment of Lower Risk Myelodysplastic Syndromes"

_hemato, doi:10.3390/hemato3010013_

Round 1

Reviewer 1 Report

Here, the author reviewed the current and emerging  treatment strategies for lower risk MDS. The review is very comprehensive , containing all current available and most emerging therapies. However I would recommend to include a short passage on aberrant inflammation, representing a relevant pathomechanism in low risk MDS and mention therapeutics  targeting the inflammasome (canakinumab/ IRAK inhibitor/NLRP3 inhibitor- currently tested in clinical trials...further preclinical data on S100A9 inhibition-ASH 21 ).

I further recommend to add a figure, summarizing current and future therapies to upgrade the review.

minor comments:

  1. abstract line 12_ please delete "it" after (ESA)
  2. line 33-> present survival longer that 1.5 years???-> you mean longer "than"?
  3. line 79 -> LR MDS implies "an" instead of "and"...
  4. line 100-> "apart" instead of "a part"
  5. line 155 -> maybe include a few words to understand the background of teh SINTRA Rev trial : A recent study, "evaluating the early treatment onset" in non...
  6. line 171: is that a full sentence : "As expected."?
  7.  line 184: Another "agent" instead of "agents"
  8. line 223/224-> "and not to be excluded those with other cytopenias"-what does it mean in this context?
  9. line 256-....-> maybe you could mention the classical criteria for the group that will benefit from immunosuppressive therapy (hypocellular, normal caryotype, blast count <5%, HLA DR15..)or at least hypocellular MDS

Author Response

First of all, I wish to thank the reviewer for the careful examination of the manuscript and the useful  comments that helped me to improve the article.

Rev1 comments

Here, the author reviewed the current and emerging  treatment strategies for lower risk MDS. The review is very comprehensive , containing all current available and most emerging therapies. However I would recommend to include a short passage on aberrant inflammation, representing a relevant pathomechanism in low risk MDS and mention therapeutics  targeting the inflammasome (canakinumab/ IRAK inhibitor/NLRP3 inhibitor- currently tested in clinical trials...further preclinical data on S100A9 inhibition-ASH 21 ). I added a brief sentence mentioning the importance of inflammation in LR MDS and therapeutic approaches targeting it. 

I further recommend to add a figure, summarizing current and future therapies to upgrade the review. I included a summary figure with targets and agents

minor comments:

  1. abstract line 12_ please delete "it" after (ESA)- corrected
  2. line 33-> present survival longer that 1.5 years???-> you mean longer "than"? corrected
  3. line 79 -> LR MDS implies "an" instead of "and"... corrected
  4. line 100-> "apart" instead of "a part" corrected
  5. line 155 -> maybe include a few words to understand the background of teh SINTRA Rev trial : A recent study, "evaluating the early treatment onset" in non... I modified the  text introducing the need to provide early treatment for MDS del5q with anemia, although not TD
  6. line 171: is that a full sentence : "As expected."? corrected
  7.  line 184: Another "agent" instead of "agents" corrected
  8. line 223/224-> "and not to be excluded those with other cytopenias"-what does it mean in this context? The language of the sentence has been changed
  9. line 256-....-> maybe you could mention the classical criteria for the group that will benefit from immunosuppressive therapy (hypocellular, normal caryotype, blast count <5%, HLA DR15..)or at least hypocellular MDS . I clarified the sentence reagarding the criteria. Some were already mentioned, but I completed the indications.

Reviewer 2 Report

This paper is a review of treatment of low risk MDS. It’s well written, extensively documented by a well-known specialist of the topic. I believe that the readers will find a great interest in this paper. I only have minor comments mainly relative to typo errors:

-              Abstract, line 12: “it” can be omitted

-              Page 4 line 171 higher and not ”. Higher”

-              Some abbreviations should be spelled out : IWG, PNH,  ATG

-              P6 line 267: FISiM was already used at the beginning of the paper

Author Response

REV 2 comments.

This paper is a review of treatment of low risk MDS. It’s well written, extensively documented by a well-known specialist of the topic. I believe that the readers will find a great interest in this paper. I only have minor comments mainly relative to typo errors:

-              Abstract, line 12: “it” can be omitted-

done

-              Page 4 line 171 higher and not ”. Higher”-

done

-              Some abbreviations should be spelled out : IWG, PNH,  ATG

done

-              P6 line 267: FISiM was already used at the beginning of the paper

done

I really wish to thank the reviewer for the careful evaluation of the text. I modified and corrected all the points indicated.